# The Potential for Cellulose Deconstruction in Fungal Genomes

Renaud Berlemont

Department of Biological Sciences, California State University Long Beach, Long Beach, CA 90840, USA;
renaud.berlemont@csulb.edu

**Definition:** Fungal cellulolytic enzymes are carbohydrate active enzymes (CAzymes) essential for the deconstruction of the plant cell wall. Cellulolytic activity is described in some glycoside hydrolases (GH-cellulases) and in auxiliary activities (AA-cellulases) families. Across environments, these enzymes are mostly produced by some fungi and some bacteria. Cellulolytic fungi secrete these enzymes to deconstruct polysaccharides into simple and easy to metabolize oligo- and mono-saccharides. The fungal ability to degrade cellulose result from their repertoire of CAZymes-encoding genes targeting many substrates (e.g., xylan, arabinose). Over the past decade, the increased number of sequenced fungal genomes allowed the sequence-based identification of many new CAZyme-encoding genes. Together, the predicted cellulolytic enzymes constitute the fungal potential for cellulose deconstruction. As not all fungi have the same genetic makeup, identifying the potential for cellulose deconstruction across different lineages can help identify the various fungal strategies to access and degrade cellulose (conserved vs. variable genomic features) and highlight the evolution of cellulase-encoding genes. Here, the potential for cellulose deconstruction identified across publicly accessible, and published, fungal genomes is discussed.

**Keywords:** fungi; cellulose; cellulase; glycoside hydrolase; LPMO; MycoCosm; CAZy





## 1. Introduction

Cellulose, composed of β-1,4-linked β-D-glucose units and constituting ~30% of the carbon in the biosphere, is the single most abundant source of organic carbon on Earth. Its synthesis (mostly by plants) and deconstruction by fungi and bacteria are some of the main drivers of global carbon cycling [1]. However, not all microbes are made equal and only some are equipped with the necessary genes encoding the cellulolytic enzymes required for cellulose deconstruction [2–4]. As complete deconstruction of the cellulose produces glucose, cellulolytic organisms and their enzymes have been the primary focus of intensive research and many biotechnological applications (e.g., biofuels production, [5–7]). In recent years, high throughput DNA sequencing and the development of bioinformatics algorithms for gene prediction and functional annotation have allowed for the gaining of insight into the diversity of cellulolytic enzymes and microbes [8–11]. The large number of sequences generated is deposited in general purpose databases (e.g., GenBank [12], Ensembl Genomes [13]) and in dedicated databases. Specifically, the MycoCosm database is a repository for sequenced fungal genomes sequenced at the Joint Genome Institute [11].

Many cellulase genes and proteins have been biochemically characterized [14]. Beside enzymes involved in the catabolism of cellulose, some characterized cellulases support other processes (e.g., plant–fungi interaction [15], cellulose production [2,16]). However, the systematic identification of cellulase genes and proteins in known cellulolytic organisms support their central function in cellulose degradation. Beside biochemically characterized genes and enzymes, the vast majority of known genes encoding cellulolytic enzymes have been identified and characterized using bioinformatic tools only [17]. Hence, the genes encoding potential cellulolytic enzymes in a given genome, in the absence of biochemical characterization, constitute the "functional potential for cellulose deconstruction" [2,18]. Describing the functional potential for cellulose deconstruction across genomes highlight

(i) the conserved evolutionary patterns in groups of organisms (i.e., phylogenetic conservatism), (ii) the adaptation to specific ecological niches (e.g., degraders vs. opportunists), and can help identify new enzymes [2,4,8,9].

Across environments, beside supporting the catabolism of cellulose from live and dead plant material, cellulases produced by fungi are required to infest and cause disease in plants and to establish symbiotic relation with plants [3,7,15,19]. Here, the potential for cellulose deconstruction across publicly accessible, and published, fungal genomes from the MycoCosm portal is discussed.

## 2. Enzymatic Deconstruction of Cellulose

To degrade cellulose, microbes have evolved distinct enzymes. The precise classification of these proteins (and many more) is presented in the carbohydrate active enzyme (CAZy) database (http://www.cazy.org accessed on 30 November 2021) [17]. Specifically, the complete cellulose breakdown requires several enzymatic activities identified in the "Glycoside Hydrolase" (GH) protein superfamily and some oxidoreductases classified in the "Auxiliary Activities" (AA) superfamily:

- Endocellulases (aka, endo-1,4-β-D-glucanase, E.C.3.2.1.4) cut randomly in the linear chain of β-1,4-linked glucose units. The complete digestion of cellulose by cellulases produces cello-oligosaccharides and cellobiose (i.e., two glucose units, $G_2$). Per the CAZy database, described protein domains endowed with endocellulase activity belong to the GH families 5, 6, 7, 8, 9, 12, 44, 45, and 48. In addition, few endocellulases have been identified in GH families 10, 51, 74, and 124. However, these last few families are poorly characterized or more frequently associated with other activities such as GH10-xylanases, GH51-arabinofuranosidase, GH74-xyloglucanase.
- Exocellulases (aka, exo-β-1,4-glucanase/cellodextrinase, E.C.3.2.1.74) digest the cellulose and cello-oligosaccharides by the extremities and release glucose ($G_1$). In addition, some exocellulases can also digest the cellobiose ($G_2$). Most described protein domains endowed with exocellulase activity belong to the GH families 5 and 9. In addition, few proteins with domains from the GH families 1 and 3 also display exocellulolytic activities. However, most GH1 and GH3 domains are known as "-osidases" degrading oligosaccharides and producing monosaccharides.
- Cellobiohydrolases (aka, E.C.3.2.1.91) hydrolyze the glucosidic linkages in cellulose and cello-oligosaccharides from their non-reducing end and release cellobiose ($G_2$). Known protein domains with cellobiohydrolases activity are from the GH families 5, 6, and 9.
- β-glucosidases (aka, E.C.3.2.1.21) hydrolyze cellobiose ($G_2$) and release glucose ($G_1$) that can be further processed thru the glycolytic pathways (e.g., Embden–Meyerhof–Parnas or Entner–Doudoroff pathways). Most known domains endowed with β-glucosidase activity belong to the GH families 1 and 3, although few have been identified in the GH families 5, 16, 30, and 39. Although β-glucosidases are not directly targeting the cellulose, these enzymes are essential for overall cellulose deconstruction. Indeed, while hydrolyzing the cellobiose ($G_2$) into glucose, β-glucosidases alleviate the cellulase inhibition by the product (i.e., $G_2$) [20–22].
- Auxiliary activities associated with cellulose deconstruction include lytic polysaccharide monooxigenases (LPMOs) from AA families 9 (formerly known as GH61) and 10 (formerly known as CBM33), per the CAZy database. Proteins in AA families 9, 10, and the more recently described AA16 (only one characterized enzyme) are copper-containing enzymes, that cleave the cellulose using an oxidative process (E.C.1.14.99.54, E.C.1.14.99.56). At the end of the reaction, the produced cellulose fragments contain a D-glucono-1,5-lactone residue at the reducing end, which spontaneously hydrolyses to an aldonic acid. In this redox process, the electrons are provided in vivo by the cytochrome-b domain of the associated "cellobiose dehydrogenase" (EC 1.1.99.18) from the AA3 family [23–25].

Based on characterized enzymes and according to the CAZy database, although some GH families are monospecific and target only one substrate, several families contain pro-

teins targeting several substrates. Monospecific cellulase families include GH7, 44, and 45. Conversely, characterized members of the GH5, 8, 9, 12, and 48 families are not always involved in cellulose deconstruction [14,17]. Specifically, only ~60% of the characterized GH5 are described as cellulases. Similarly, ~75% of GH12, ~96% of GH9, and ~75% of GH48 target the cellulose. The non-cellulolytic enzymes from these GH families include some GH5-mannases, some licheninases (GH5, GH9, GH12), some xylanases (GH5, GH8, GH12), and some chitinases (GH48), among others.

Next, although some GH families are "monospecific", some characterized proteins in monospecific families display distinct mode of actions. For example, in the GH6 family, ~65% of the characterized enzymes are described as cellobiohydrolases (E.C.3.2.1.91) and ~37% as endocellulases (E.C.3.2.1.4). Finally, few GH families contain characterized proteins active on multiple substrates and/or displaying multiple modes of action.

Finally, regarding the characterized AA enzymes, all the characterized LPMO-AA9 target the cellulose whereas only ~50% and ~35% of the characterized LPMO-AA10 and LPMO-AA3 target the cellulose, respectively.

Although GHs and AAs are generally identified as single domain proteins, they are frequently associated with other catalytic or non-catalytic domains in composite multi-domain proteins (MDGHs and MDAAs) [3–5]. Proteins with several catalytic domains include proteins with several GH domains or some non-GH domains (e.g., esterase) [9,26]. Carbohydrate-binding modules (CBMs) are non-catalytic domains frequently associated with GH domains. The presence of CBMs enhances the enzyme–substrate interaction by anchoring the catalytic domain to the substrate [27]. The anchoring reduces diffusion from the substrate and locally increases the concentration of catalytic domains, thus improving the overall polysaccharide degradation [27,28].

Hereafter, the predicted genes and corresponding enzymes from the GH families 5, 6, 7, 8, 9, 12, 44, 45, and 48 will be referred to as the "GH-cellulases" whereas the predicted GH families 1 and 3 will be referred to as the β-glucosidases. Similarly, the predicted genes and enzymes from the AA families 9, 10, and 16 will be referred to as the "AA-cellulases".

## 3. MycoCosm and Data Availability

In the past few decades, high-throughput DNA sequencing techniques and the associated bioinformatic approaches (e.g., genome assembly, gene prediction) have allowed the sequencing and characterization of many microbial genomes. Specifically, the MycoCosm project at the Joint Genome Institute (https://mycocosm.jgi.doe.gov/mycocosm/home, accessed on 30 November 2021) [11,29] aims to (i) better understand the plant–fungi interaction (e.g., phytopathogens and mycorrhizal symbionts), (ii) provide new insight into the conversion of biopolymer (e.g., plant cell wall biorefinery), and (iii) mine the potential of the yet undiscovered natural arsenal of potential applications (e.g., antibiotic).

The presented data was directly retrieved from the MycoCosm portal as of November 2021 and only publicly accessible and published data are included. All the abbreviation and enzyme names are consistent with the publicly accessible data. When a specific genome is mentioned, the corresponding reference is provided. However, readers are encouraged to check the MycoCosm portal to retrieve additional information and references.

## 4. Cellulases Distribution across Fungal Phyla

Most analyzed fungal genomes contained many genes encoding potential enzymes for cellulose deconstruction (Figure 1). Across lineages, larger genomes (expressed in total predicted genes) had a higher number of potential genes for cellulose deconstruction (Figure 1A,D). In fungi, larger genomes reflect some duplication events, some remnant of polyploidization events, and some horizontal genes transfers [30–32]. This suggests that cellulose deconstruction is a core function and that evolution towards increased cellulolytic potential is central to the evolution of fungi (Figure 1). However, these trends were not supported across the genomes from the less abundant phylum Cryptomycota (*n* = 2 genomes), Mucoromycota (*n* = 24), Microsporidia (*n* = 8), and Zoopagomycota

($n$ = 7). In these mostly parasitic groups of fungi, genome reduction removing the unnecessary genes led to smaller genomes [32–34]. These observations need to be confirmed when more species of these phyla are sequenced.

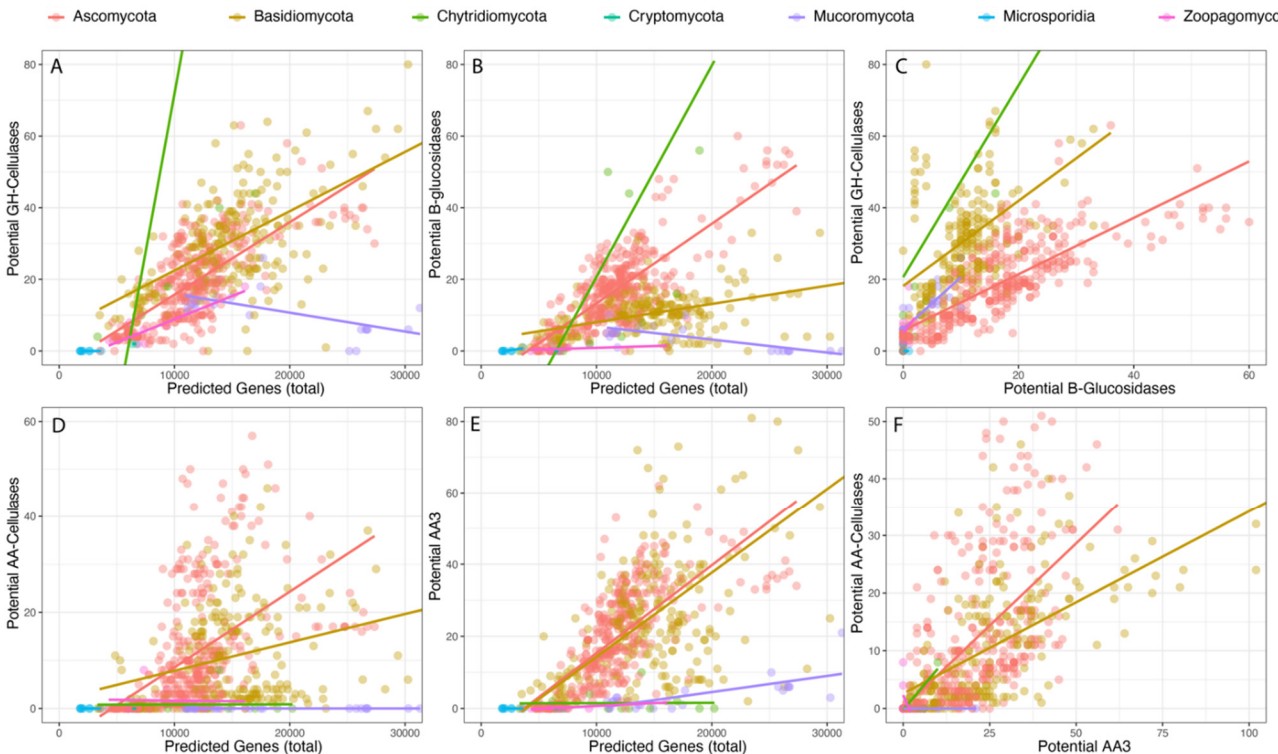

**Figure 1.** The total number of predicted genes encoding potential GH-cellulases (**A**), β-glucosidases (**B**), AA-cellulases (**D**), and AA3 (**E**) enzymes across sequenced fungal genomes and the relation between the number of cellulolytic enzymes (GH-cellulases and AA-cellulases) and the number of enzymes active on cellobiose (**C,F**). Lines depict the linear regression between the number of predicted genes encoding the potential enzymes and the predicted total number of genes.

Interestingly, in the abundant Ascomycota ($n$ = 374) and Basidiomycota ($n$ = 219), most genomes contained more predicted GH-cellulases than β-glucosidases. Several genomes had few predicted GH-cellulase and no β-glucosidases, whereas a reduced number of genomes contained no predicted GH-cellulases but few β-glucosidases (Figure 1C). This suggests that some members of the Ascomycota and Basidiomycota phyla, with a reduced number of predicted enzymes for cellulose deconstruction, are not actively degrading the plant cellulose or use different mechanisms to produce cellulolytic enzymes (e.g., gene overexpression). The same trends were observed for the auxiliary activities (Figure 1D–F). However, unlike for GHs, many genomes had a reduced number of genes for predicted cellobiose dehydrogenases (i.e., AA3) (Figure 1F). Having no genes for the polysaccharide deconstruction but genes supporting the processing of the intermediate deconstruction product has been described as a strategy in opportunistic microbes taking advantage of the active cellulose degraders [2]. However, globally, most fungal genomes contained many predicted genes/enzymes supporting cellulose deconstruction. Having several—generally coregulated—genes for cellulases increases the overall cellulolytic activity [35–38]. In addition, characterized cellulolytic enzymes derived from the same genome tend to display slightly distinct substrate specificity and kinetic parameters [7,30,38]. In this context, producing many enzymes with varying specificity supports the optimal overall cellulose deconstruction.

When focusing on major fungal lineages, not all the taxa shared the same trends. Among others, first, Mucoromycota and Zoopagomycota appeared to have extremely

reduced potential for cellulose deconstruction. Second, although the frequency of potential cellulases was similar across Basidiomycota and Ascomycota, the frequency of potential β-glucosidases and auxiliary activities was significantly higher in Ascomycota. Finally, although the few analyzed genomes from the Chytridiomycota contained a very high number of predicted GH-cellulases, especially in Neocallimastigomycetes ($n = 5$), only a few AAs were identified in this phylum. Together these observations highlight the variation in the gene content supporting the cellulose deconstruction in sequenced fungal genomes. Hereafter, the potential for cellulose deconstruction in the abundant genomes from the Ascomycota and Basidiomycota phyla is discussed.

## 5. Cellulases Distribution across Basidiomycota

Among the Basidiomycota ($n = 219$), the class Agaricomycetes ($n = 172$) is the most abundant. In this class, beside few analyzed genomes (i.e., Gansp1 [39], Lacbi2 [40], Schco3 [41], and Phlbr1 [39]), all the genomes had a combination of potential cellulases, β-glucosidases, and auxiliary activities (Figure 2). On average, 0.29% of the total predicted genes in Agaricomycetes encode potential cellulases. However, although few genomes lacked potential cellulases, up to 0.68% of the predicted genes encode potential cellulases.

First, in the order Agaricales ($n = 50$), analyzed genome ranged from 5420 to 49,694 total predicted genes and potential cellulase accounted for 0.35–0.14% of the predicted genes. In this order, the number of predicted cellulases correlated with the total number of predicted genes ($r_{Pearson} = 0.40$, $p < 0.001$). Next, Polyporales ($n = 47$), with genome ranging from 9113 to 18,244 total predicted genes and potential cellulases accounting for up to $0.31 \pm 0.10\%$ of the predicted genes, included one genome with no predicted cellulase (i.e., *Ganoderma* sp. 10,597 SS1 V1.0–Gansp1) [39]. As described for the Agaricales, the number of predicted cellulases in Polyporales correlated with the total number of predicted genes ($r_{Pearson} = 0.31$, $p < 0.001$). Next, Cantharellales ($n = 6$) have genome ranging from 15,157 to 19,659 predicted genes with $0.45 \pm 0.22\%$ being predicted cellulases. Although relatively enriched in predicted cellulases, the number of predicted genes did not mirror the variation in the genome size of Cantharellales. Finally, most Boletales ($n = 40$), with genome ranging from 9409 to 21,458 total predicted genes, had relatively few genes encoding potential cellulase ($0.17 \pm 0.06\%$). In addition, as described for the Cantharellales, the variation in the number of genes encoding potential cellulases in Boletales did not correlate with the genome size.

Testing the correlation between the number of predicted cellulases and the total number of genes across genomes is a proxy to investigate whether genes are under selection during the genome expansion. When the number of predicted cellulases and the total number of predicted genes correlate this suggests that, in the analyzed group, these genes and the function they encode are under positive selection: the genome expansion involves the potential for cellulose deconstruction. Conversely, when these values do not correlate it suggests that the genes are not under selection. Alternatively, the number of predicted cellulases and the total number of genes can appear unrelated if the analyzed genomes all have a similar, but not identical, number of predicted genes and cellulases, or if the analyzed genomes is not representative of the taxon's diversity, among other reasons.

In Agaricomycetes, the most abundant potential cellulases detected were GH5 (from 1 to 76 identified genes per genome and 22.1 identified genes on average) and AA9 (from 1 to 52 identified genes per genome and 12.8 identified genes on average), after removing Gansp1 (with no predicted cellulase). In addition to being the prevalent cellulases, GH5 was also the most consistent across the Agaricomycetes genomes (coefficient of variation, CoV = 0.38). Next to these enzymes, the most abundant predicted cellulases were GH7, GH12, and GH45, with an average number of 3.28, 2.46, and 2.08 predicted genes per genome, respectively. Finally, few GH6, GH9, GH44, AA10, and AA16 were identified, in some genomes only, whereas no GH8 nor GH48 were identified in Agaricomycetes. Finally, even after dropping the Gansp1 genome, no cellulase family was found consistently in all the analyzed Agaricomycetes genomes.

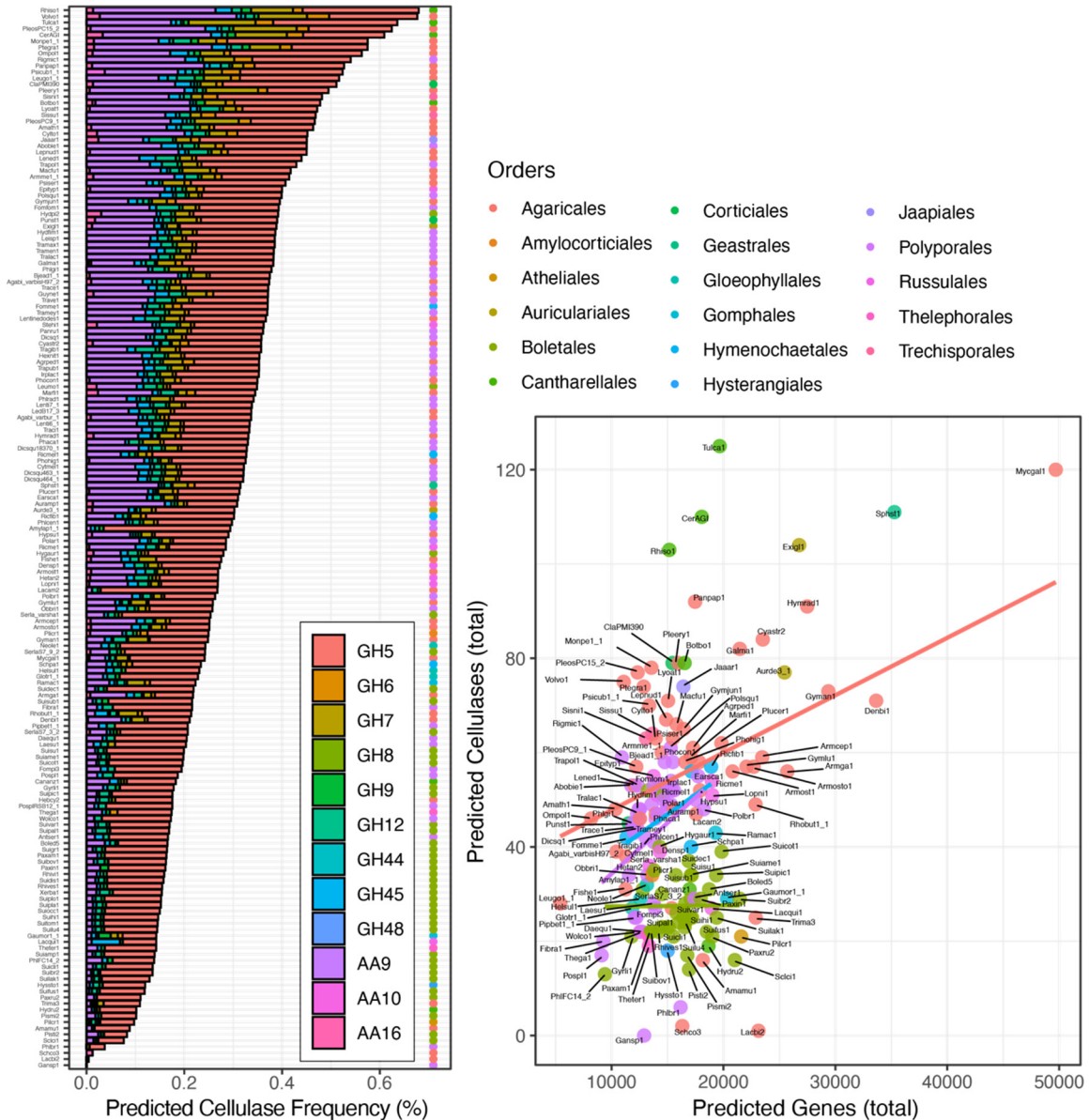

**Figure 2.** The functional potential for cellulose deconstruction in sequenced genomes from the class Agaricomycetes (*n* = 172 genomes). Frequency of the predicted potential cellulases (left) and total number of predicted potential cellulases across orders of Agaricomycetes (right). Lines depict the linear regression between the predicted cellulases and the predicted number of genes, if correlated (i.e., P$_{Pearson}$ < 0.05).

The 47 non-Agaricomycetes Basidiomycota were unevenly distributed among few classes including Pucciniomycetes (*n* = 13), Tremellomycetes (*n* = 9), Exobasidiomycetes (*n* = 7), and Ustilaginomycetes (*n* = 6), plus some very few genomes from other classes (data not shown). Overall, the lack of available data limited the discussion of the potential for cellulose deconstruction in these lineages. However, very briefly, all the analyzed non-Agaricomycetes Basidiomycota had predicted cellulases. GH5, found in all the genomes, were the most abundant predicted enzymes. Many lineages lacked predicted auxiliary activities whereas most Pucciniomycetes and the Wallemiomycetes had some AA9. Specifically, most Pucciniomycetes had a broad repertoire of predicted cellulases from GH families 5, 7, 12, 45, and AA9, together accounting for 0.19 to 0.35% of the total predicted genes. Next, ~0.38% of the predicted genes in the two Wallemiomycetes genomes [42,43] (~5000 predicted genes) were predicted cellulases from GH5 or AA9 only, accounting for

~46% and 54% of the predicted cellulases, respectively. Finally, in the unique genome from the class Mixiomycetes (i.e., *Mixia osmundae* IAM 14,324 v1.0–Mixos1) [44], 28 of the 6903 total predicted genes, were cellulases from GH5 (14), GH12 (12), and GH45 (2).

## 6. Cellulases Distribution across Ascomycota

Among the Ascomycota (*n* = 374), genomes from the classes Saccharomycetes (*n* = 43), Sordariomycetes (*n* = 85), Eurotiomycetes (*n* = 107), Dothideomycetes (*n* = 93), were the most abundant. Saccharomycetes displayed reduced potential for cellulose deconstruction and contained some potential GH5-cellulases and few potential GH45 in *Ascoidea rubescens* NRRL Y17699 (Ascru1) [45] and *Ambrosiozyma philentoma* NRRL Y-7523 (Ambph1) [45]. Finally, *Lipomyces starkeyi* NRRL Y-11557 (Lipst1_1) [45] was the only Saccharomycetes having a predicted potential cellulolytic AA9 (data not shown).

In Sordariomycetes (Figure 3), predicted genes encoding cellulases accounted for 0.06% to 0.78% of the total predicted genes suggesting that members of this class have adopted various ecological strategies regarding cellulose deconstruction. Specifically, first, Xylariales (*n* = 8), Glomerellales (*n* = 6), and Magnaporthales (*n* = 3), with only few genomes ranging from 9411 genes to 15,413 genes, each displayed high frequency of genes encoding potential cellulases (Figure 3). Next, Hypocreales (*n* = 58), with genomes ranging from 6451 to 27,347 predicted genes, had generally lower potential for cellulose deconstruction. In this group, the number of predicted cellulolytic enzymes was strongly correlated to the total number of predicted genes ($r_{Pearson}$ = 0.71, *p* < 0.001, Figure 3). More specifically, although most *Nectriaceae* (*n* = 20), had significantly larger genomes than *Hypocreaceae* (*n* = 20), *Clavicipitaceae* (*n* = 7), and *Ophiocordycipitaceae* (*n* = 5), suggesting distinct life-styles, the relation between number of predicted cellulase and the total number of genes enzyme remained mostly unchanged for these four families.

In Sordariomycetes, most identified genes encoding potential cellulases were GH5 (from 3 to 24 genes/genome) and GH7 (from 0 to 8) although other GH-cellulases were also identified. Interestingly, the frequency of these GH-encoding genes remained mostly unchanged in most Sordariomycetes regardless of the overall variation in the genome size. On the contrary, the frequency of the abundant predicted AA9 cellulases (from 1 to 46 fluctuated extensively. In addition, few potential AA16 cellulases were identified in ~40% of the sequenced genomes. The overall frequency of predicted AA-cellulases fluctuated across genomes and was relatively high in most Glomerellales (*n* = 6), Xylariales (*n* = 8), and Magnaporthales (*n* = 3) and low in most Sordariales (*n* = 6) and Hypocreales (*n* = 58), except for the two only members of the *Stachybotryaceae* family: *Stachybotrys chartarum* IBT 40,288 (Stach1) and *S. chlorohalonata* IBT 40,285 (Stachl1) [46]. The systematic presence of multiple potential cellulolytic enzymes, AA, and GH-cellulases in Sordariomycetes suggest the importance of the cellulose deconstruction for this group. In addition, as the increasing number of potential GH-cellulases mirrors the variation in the overall genome size, this suggests that these cellulases are under strong positive selection, at least in some lineages (e.g., Glomerellales and Xylariales).

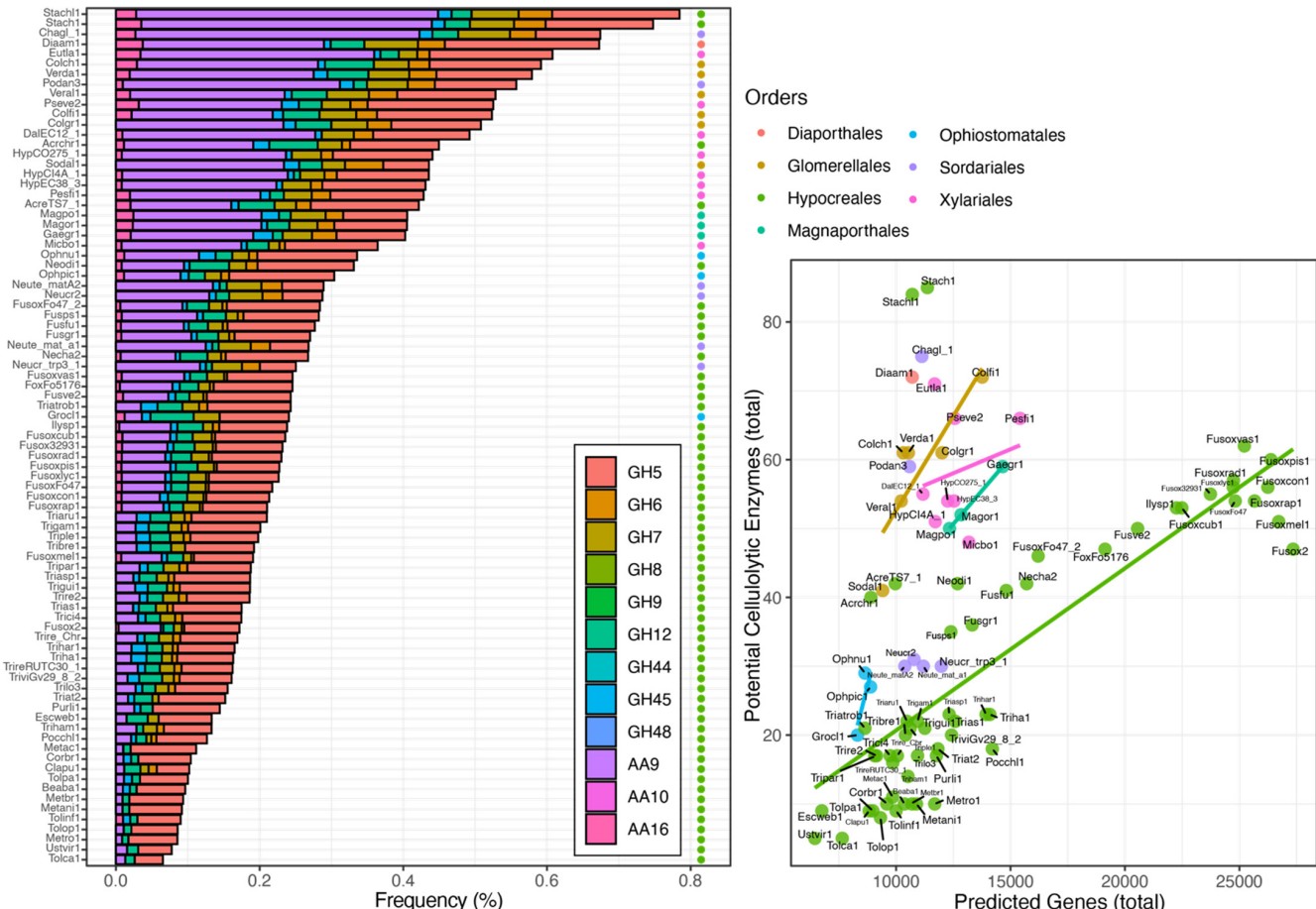

**Figure 3.** The functional potential for cellulose deconstruction in sequenced genomes from the class Sordariomycetes (*n* = 85 genomes). Frequency of the predicted potential cellulases (left) and total number of predicted potential cellulases across orders of Sordariomycetes (right). Lines depict the linear regression between the predicted cellulases and the predicted number of genes, if correlated (i.e., $P_{Pearson}$ < 0.05).

In the Eurotiomycetes (Figure 4), with genomes ranging from 6907 to 15,139 predicted genes, genes encoding potential cellulases accounted for < 0.05% to 0.4%. Members of the Eurotiales order (*n* = 81) were the only genomes, although highly variable, to show a slight increase in the number of predicted gene encoding potential cellulases as the genomes contain more genes. More specifically, the number of potential cellulases and the total number of predicted genes correlated only in the *Aspergillaceae* family (*n* = 75 genomes, $r_{Pearson}$ = 0.44, *p* < 0.001). In the other orders of Eurotiomycetes, such as Chaetothyriales (*n* = 15), Verrucariales (*n* = 2), and Onygenales (*n* = 8), the number of genes encoding potential cellulolytic enzyme was mostly unaffected by the overall number of predicted genes. Interestingly, *Phaeomoniella chlamydospora* UCRPC4 (Phach1) [47], the only Phaeomoniellales analyzed, had many genes encoding potential cellulolytic activities despite having the smallest genomes in the Eurotiomycetes group.

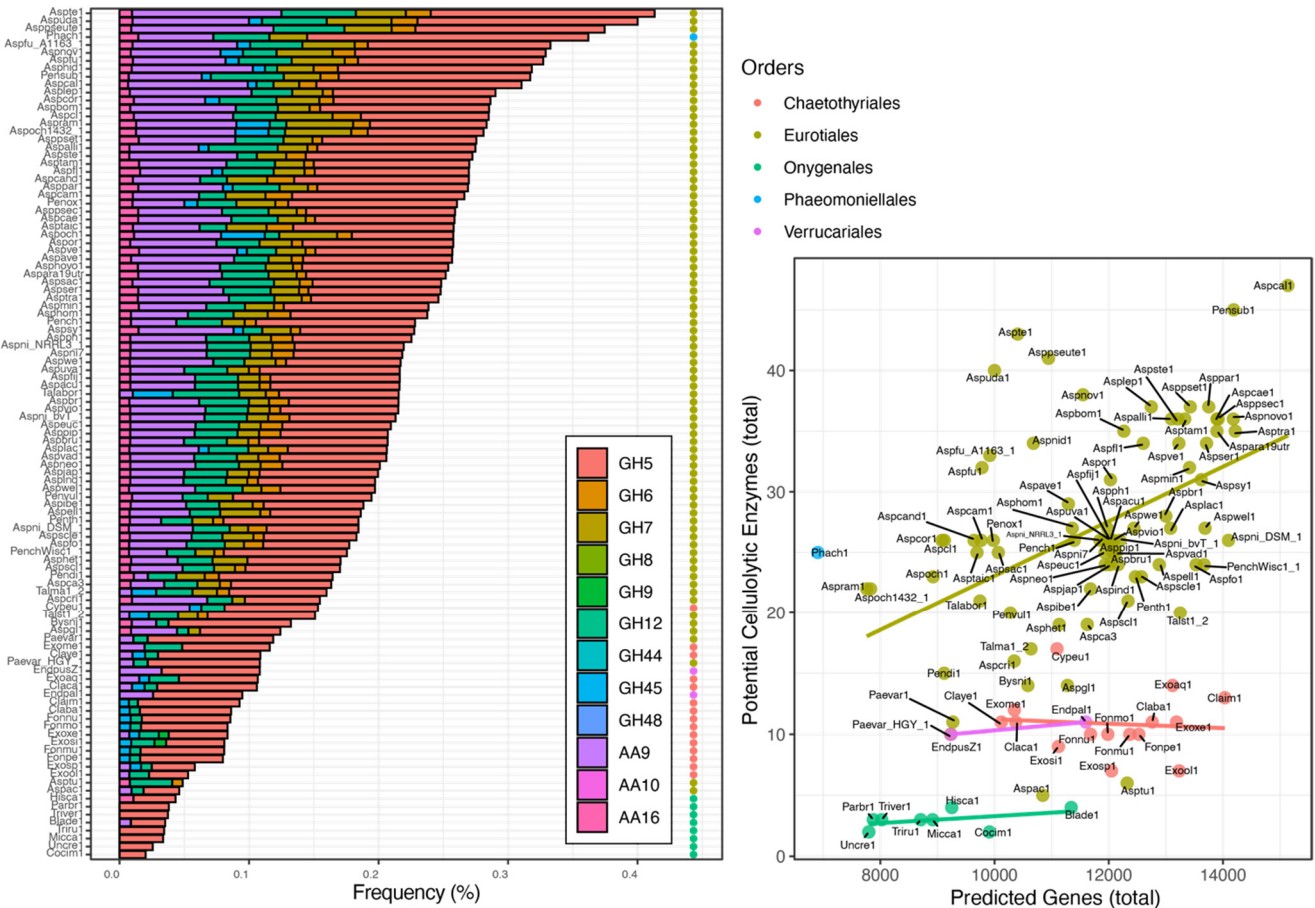

**Figure 4.** The functional potential for cellulose deconstruction in sequenced genomes from the class Eurotiomycetes (*n* = 107). Frequency of the predicted potential cellulases (left) and total number of predicted potential cellulases across orders of Eurotiomycetes (right). Lines depict the linear regression between the predicted cellulases and the predicted number of genes, if correlated (i.e., $P_{Pearson}$ < 0.05).

Genes encoding potential cellulolytic GHs and AAs were identified in all the Eurotiomycetes. Potential GH5 cellulases were identified in all the genomes apart from *Aspergillus tubingensis* v1.0 (Asptu1) [48]. Potentially cellulolytic GH7 and GH12 were also abundant in many genomes although some taxa completely lacked these genes. Regarding the auxiliary activities, although most analyzed genomes contained many potential AA9 and few AA16, most members of the Onygenales and Chaetothyriales families completely lacked these genes.

Among the Dothideomycetes (Figure 5), genomes ranged from 7572 to 21,730 predicted genes and potential cellulases accounted from 0% to 0.68%. The genome of *Microthyrium microscopicum* CBS 115,976 v1.0 (Micmi1) [49] contained no identified gene encoding potential cellulolytic activity. Next, *Polychaeton citri* v1.0 (Polci1) [49] and *Piedraia hortae* CBS 480.64 v1.1 (Pieho1_1) [49], the only two sequenced members of the Capnodiales family, had a only a few GH5. All the other genomes displayed a combination of genes encoding potential GH and AA targeting the cellulose.

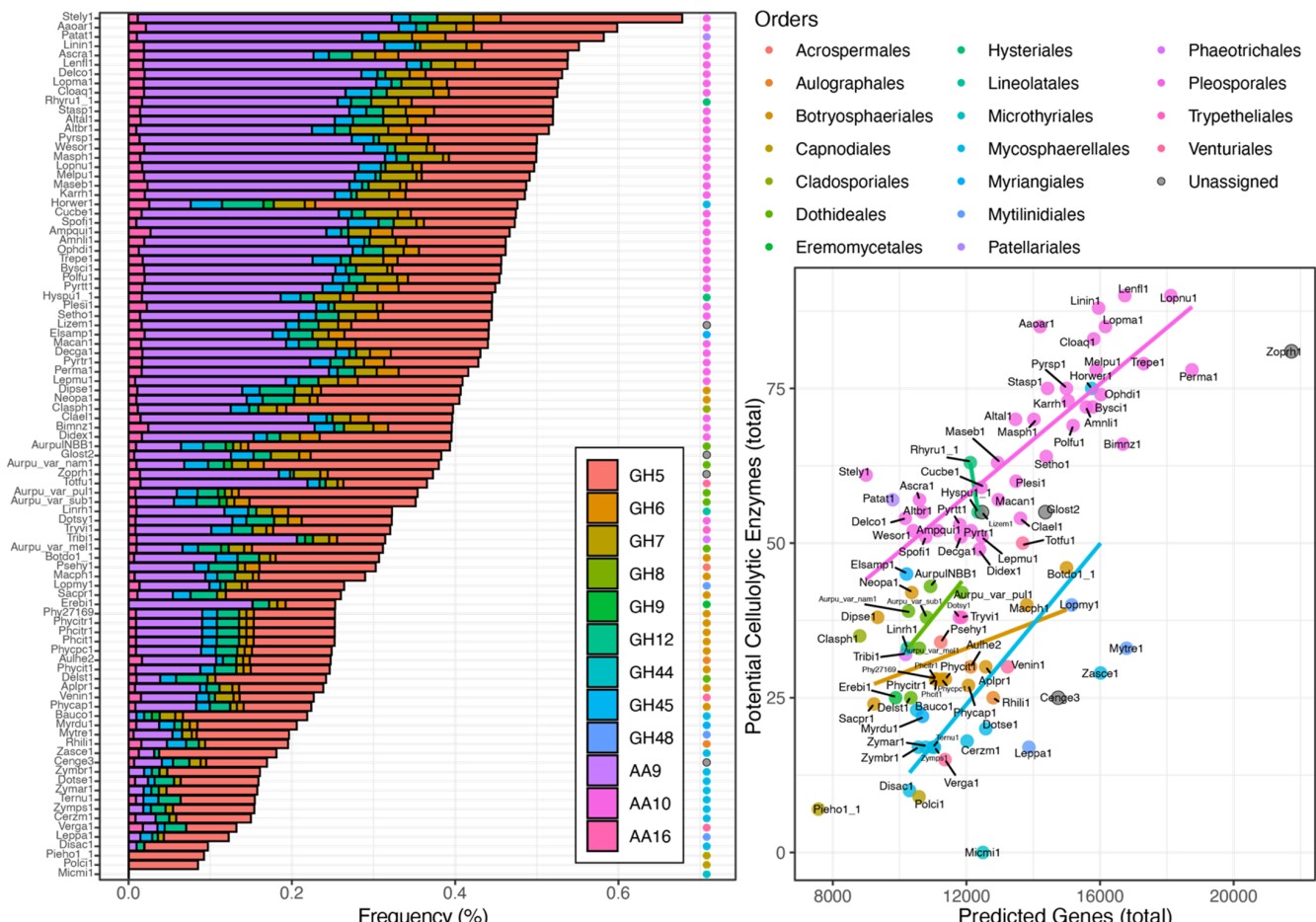

**Figure 5.** The functional potential for cellulose deconstruction in sequenced genomes from the class Dothideomycetes (*n* = 93). Frequency of the predicted potential cellulases (left) and total number of predicted potential cellulases across orders of Dothideomycetes (right). Lines depict the linear regression between the predicted cellulases and the predicted number of genes, if correlated (i.e., $P_{Pearson} < 0.05$).

Focusing on lineages with multiple genomes, most Mycosphaerellales (*n* = 10) and Botryosphaeriales (*n* = 13) displayed reduced potential for cellulose deconstruction whereas most Dothideales (*n* = 6) and Pleosporales (*n* = 38) had many potential enzymes targeting the cellulose. Some genes for potential cellulolytic GH5, 6, 7, 9, 12, and 45 were identified in all Dothideales except for *Delphinella strobiligena* CBS 735.71 v1.0 (Delst1) [49], lacking both genes encoding GH6 and GH9. Next, AA9 and GH5 and some less abundant GH6, 7, 12, 45, and AA16 were identified in all Pleosporales. A similar trend was observed in Botryosphaeriales although predicted GH5-cellulases were generally more abundant than predicted AA9. In addition, in this group, *Saccharata proteae* CBS 121,410 v1.0 (Sacpr1) [49] lacked potential GH6. Finally, members of the Mycosphaerellales order had low potential for cellulose deconstruction (i.e., 0.2% of the predicted genes on average), with many genomes lacking genes for the aforementioned predicted enzymes. Although the potential for cellulose deconstruction varied extensively in these orders, the number of predicted genes for cellulases correlated with the total number of predicted genes in Pleosporales ($r_{Pearson}$ = 0.81), Mycosphaerellales ($r_{Pearson}$ = 0.75), and Botryosphaeriales ($r_{Pearson}$ = 0.48, all *p*-values < 0.001).

## 7. Cellulases Distribution across the Other Phyla

Considering the other phyla (Figure 6), although more genomes existed, the publicly released sequenced genomes consisted in a reduced number of Mucoromycota (*n* = 24), Chytridiomycota (*n* = 12), and Zoopagomycota (*n* = 7). Except for the Neocallimastigomycetes (*n* = 5) in the phylum Chytridiomycota, most of these genomes contained reduced number of GH-cellulases and sometime few AA-cellulases. In these genomes with reduced potential for cellulose utilization, GH5 and GH9 genes were the most abundant genes encoding potential cellulolytic enzymes.

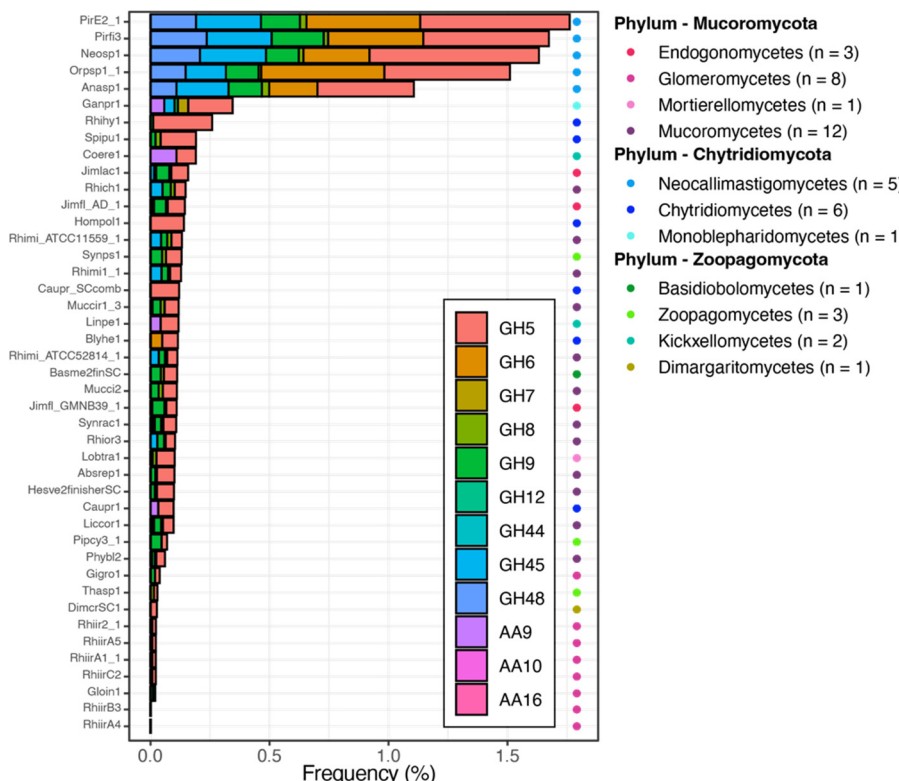

**Figure 6.** Frequency of the predicted potential cellulases across genomes of the less abundant fungal phyla.

Neocallimastigomycetes displayed small genomes ranging from 10,992 to 20,219 predicted genes and high frequency, from ~1.11 to ~1.76%, of genes encoding potential cellulolytic enzymes. In addition, compared to most of the other lineages analyzed, Neocallimastigomycetes contained, next to abundant GH5 and GH6, a very unique set of GH8, 9, 45, and 48, whereas no gene encoding GH7, GH12, and GH44 nor any cellulolytic AA was detected.

## 8. Conclusions and Prospects

The identification of genes encoding the cellulolytic enzymes in sequenced fungal genomes is instrumental to understanding how fungi degrade plant material and support carbon cycling across environments. The MycoCosm portal centralizes the information about fungal genomes sequenced at the Joint Genome Institute and provides standardized information (e.g., predicted genes, functional annotation, etc.).

Most sequenced fungal genomes presented here contain predicted cellulases. This supports the importance of the plant cell wall degradation, and of the interaction with the plant cell wall, for fungi. This also highlights how essential fungi are for carbon cycling. Across lineages, GH5 are generally the most abundant and consistent predicted cellulases. The almost exclusively fungal GH7-cellulases are generally the second most abundant GH-

cellulases, whereas the other predicted GH-cellulases are generally less abundant but their distribution within fungal families is generally conserved. Regarding the predicted AA-cellulases, these enzymes are identified in most lineages, especially AA9. However, in some groups (e.g., Pleosporales, Sordariales, Xylariales) the number of predicted AA-cellulases is higher than the number of predicted GH-cellulases. Conversely, some genomes associated with many predicted enzymes for cellulose deconstruction (e.g., Neocallimastigomycetes) had no detected AA-cellulases.

Increasing the number and diversity of cellulase genes is one strategy used by microbes to support cellulose deconstruction [2,3,18]. However, providing a comprehensive understanding of how microbes degrade cellulose requires more than just identifying their functional potential. Some essential approaches include elucidating how (the level of expression) and when (condition for expression) cellulase genes are being expressed (i.e., transcriptomic—e.g., [37,38,43,50]) and cellulolytic enzymes secreted (i.e., proteomic—e.g., [51,52]). Finally, integrating these data with the substrate specificity, kinetics, and stability of individually characterized cellulases (e.g., [6,53,54]) will provide the necessary information to describe the cellulose deconstruction.

Although the development of standardized transcriptomic and proteomic approaches aimed at elucidating the carbohydrate metabolism in fungi are emerging (e.g., [50]), genome sequencing is becoming more accessible to scientists and new genomes, including genomes from the less characterized groups, are released frequently [11]. As of November 2021, many unpublished genomes were already available on the MycoCosm portal [11]. These genomes were not included here to let the authors publish primary literature. These genomes included >90 Ascomycota (e.g., 51, Sordariomycetes, 23 Dothideomycetes, 12 Eurotiomycetes), 48 Basidiomycota (i.e., 47 Agaricomycetes and 1 Microbotryomycetes), 4 Mucoromycota, and 1 Chytriodiomycota. The uneven distribution of the genomes to be released mirrors the biased diversity of the available genomes and the abundance and diversity of fungal lineages across environments. In the future, including these recently sequenced genomes, and hopefully more from poorly sampled lineages, will provide new opportunities to investigate how fungi deconstruct plant cellulose.

**Funding:** This research received no external funding.

**Institutional Review Board Statement:** Not applicable.

**Informed Consent Statement:** Not applicable.

**Data Availability Statement:** Data presented in this entry are all publicly available on the Joint Genome Institute (JGI) website: https://mycocosm.jgi.doe.gov/mycocosm/home (accessed on 30 November 2021).

**Acknowledgments:** The sequence data were annotated and made publicly available by the US Department of Energy Joint Genome Institute (http://www.jgi.doe.gov/), in collaboration with the user community. I thank all investigators who contributed to the submission and sequencing of the fungal genomes.

**Conflicts of Interest:** The authors declare no conflict of interest.

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
