# Peer review of "The Potential for Cellulose Deconstruction in Fungal Genomes"

_encyclopedia, doi:10.3390/encyclopedia2020065_

Round 1
Reviewer 1 Report
Dear Author
I checked your Entry. It is great. Congratulations.
Regards
Reviewer
Author Response
Thank you!
Reviewer 2 Report
As the plant biomass is chemically complex, the organisms, capable to degrade it, developed various strategies to successfully decompose polysaccharides and to cope with lignin. Among great number of species, fungi are very interesting group and brought the researchers attention in recent years. Taken together with rapid development of high-throughput sequencing techniques it allows to elucidate new approach to fungal biology and wood decomposition processes. There is still lack of papers dealing with complex view on fungal genetics on wood degradation in relation to evolution and future prospects. Consequently, the presented paper is of high importance as it gives the best possible knowledge about it. However, in the opinion of the reviewer the paper needs some corrections listed below:
Major:
- There is lacking a research hypotheses in the last paragraph of introduction. Moreover, reading such amount of results one could expect a good discussion, and I must admit that I am disappointed with. There should be more dealing with fungal evolution, ecological types (WRF, BRF), transcriptomic and proteomic data.
- The author concludes that ability to decompose cellulose comes with number of genes coding for cellulases. But it is only small part of whole picture. What about gene expression? We do not know if these genes are expressed, they might be disrupted. There is lack of information how their expression is regulated, which is complex process. We do not know about the activity of these cellulases (enzymes). Therefore, the cellulolytic potential of this fungal species is hard to estimate based on number of genes coding for cellulases.
- Are there any proof of genome duplication in these fungi? This should be discussed.
Minor:
Line 7 – “Fungal cellulolytic enzymes are carbohydrate active enzymes (CAzymes) essential for the deconstruction of the plant cell wall.” – CAZy is the name of database. I am afraid there is no such a name in enzymology. It should be rather referred to EC.
Line 12 “The fungal ability to degrade cellulose result from their repertoire of CAZymes- encoding genes targeting many substrates (e.g., xylan, chitin).” – How ability do decompose cellulose may result from chitinases? – Please explain.
Line 13 – “ Over the past decade, the increased number of sequenced fungal genomes allowed the sequence-based identification of many new CAZyme-encoding genes.” – Which genes? Please, be specific.
Line 24 – “ The polysaccharide cellulose…” – please delete “polysaccharide”
Line 40 – “ Beside characterized genes and enzymes, the vast majority of known genes encoding cellulolytic enzymes have been identified and characterized in silico only” – I suppose I know what do you mean, but I thing that these sentence needs rephrasing to clarify.
Line 49 – “Across environments, fungi produce cellulases to (i) infest and cause disease in plants, (ii) to establish symbiotic relation with plants, and (iii) to support the catabolism of cellulose to produce energy” – what about saprotrophic fungi?
Line 67 – “However, these last few families are more frequently associated with other activities” – which activities?
Line 149 – “This suggests that cellulose deconstruction is a core function and that evolution towards increased cellulolytic potential is central to the evolution of fungi” – What about that parasitic organisms have decreasing genomes due to their increasing metabolic specialization? Could you refer your statements more to evolution data?
Figures – Please correct axis names. I suppose that it should be for example “Number of genes coding for cellulases”. Please, also check across the manuscript (f. e. line 160).
Line 163 – “This suggests that some members of the Ascomycota and Basidiomycota phyla, with reduced number of predicted enzymes for cellulose deconstruction, are not actively degrading the plant cellulose.” – Are you sure? What about gene expression regulation? What about enzyme kinetics?
Line 166 – “However, unlike for GHs, many genomes had only few genes for potential cellobiose dehydrogenases (i.e., AA3)” – or even one.
Line 171 – “Having several -generally coregulated- genes for cellulases increases the overall cellulolytic activity.” - I suppose I know what do you mean, but I thing that these sentence needs rephrasing to clarify.
Line 189 – Please, give full fungal names. Please, check across the manuscript.
Author Response
As the plant biomass is chemically complex, the organisms, capable to degrade it, developed various strategies to successfully decompose polysaccharides and to cope with lignin. Among great number of species, fungi are very interesting group and brought the researchers attention in recent years. Taken together with rapid development of high-throughput sequencing techniques it allows to elucidate new approach to fungal biology and wood decomposition processes. There is still lack of papers dealing with complex view on fungal genetics on wood degradation in relation to evolution and future prospects. Consequently, the presented paper is of high importance as it gives the best possible knowledge about it. However, in the opinion of the reviewer the paper needs some corrections listed below.
Thank you.
Major:
- There is lacking a research hypotheses in the last paragraph of introduction. Moreover, reading such amount of results one could expect a good discussion, and I must admit that I am disappointed with. There should be more dealing with fungal evolution, ecological types (WRF, BRF), transcriptomic and proteomic data.
This manuscript is an "entry" paper in the MDPI-Encyclopedia journal. Per the editorial guidelines, "entry" type papers, are descriptions of mature knowledge for reference by researchers, it is not meant to provide, nor to answer, any new hypothesis.
- The author concludes that ability to decompose cellulose comes with number of genes coding for cellulases. But it is only small part of whole picture. What about gene expression? We do not know if these genes are expressed, they might be disrupted. There is lack of information how their expression is regulated, which is complex process. We do not know about the activity of these cellulases (enzymes). Therefore, the cellulolytic potential of this fungal species is hard to estimate based on number of genes coding for cellulases.
This is correct. The analysis of the genome sequence provides a limited understanding of the whole picture, and the manuscript is very clear about that. The papers is all about the predicted functional potential deduced from identified sequences. However, one paragraph was added to highlight the need for more "integrated" study combining transcriptomic, proteomic and biochemical characterization of the cellulolytic fungi, under different environmental conditions (LL396-408).
- Are there any proof of genome duplication in these fungi? This should be discussed.
I understand this concern and provided several references pointing at evolutionary mechanisms leading to gene count fluctuation in fungal genomes. However, these mechanisms (e.g., duplication, polyploidy, genome streamlining) are not specific to fungi nor to carbohydrate active enzymes and I am not actively investigating this aspect in this entry.
Minor:
Line 7 – “Fungal cellulolytic enzymes are carbohydrate active enzymes (CAzymes) essential for the deconstruction of the plant cell wall.” – CAZy is the name of database. I am afraid there is no such a name in enzymology. It should be rather referred to EC.
The classification of proteins is often based on 3D-structure (when available) and sequence homology - not the activity (EC number). Interestingly, the standing databases (e.g., PFam, Interpro) refer to and often contain hyperlink to the CAZy database when it comes to "carbohydrate active enzymes" thus making it the "de facto" classification of these enzymes.
Line 12 “The fungal ability to degrade cellulose result from their repertoire of CAZymes- encoding genes targeting many substrates (e.g., xylan, chitin).” – How ability do decompose cellulose may result from chitinases? – Please explain.
Your are correct! This was a bad example. The word chitin has been replaced by "arabinose".
Line 13 – “ Over the past decade, the increased number of sequenced fungal genomes allowed the sequence-based identification of many new CAZyme-encoding genes.” – Which genes? Please, be specific.
I am not sure to understand this comment. I believe that the abstract summarizes the main text and should not contain too many details. However, to clarify, these "new CAZyme-encoding genes" are all the genes that have been identified thanks to high throughput sequencing (literally thousands of genes) . It is not about new categories of genes/protein families although some new families have been identified - but this is not what I am saying.
Line 24 – “ The polysaccharide cellulose…” – please delete “polysaccharide”
Modified as suggested
Line 40 – “ Beside characterized genes and enzymes, the vast majority of known genes encoding cellulolytic enzymes have been identified and characterized in silico only” – I suppose I know what do you mean, but I thing that these sentence needs rephrasing to clarify.
The sentence was modified for clarification: " Beside biochemically characterized genes and enzymes, the vast majority of known genes encoding cellulolytic enzymes have been identified and characterized using bioinformatic tools only"
Line 49 – “Across environments, fungi produce cellulases to (i) infest and cause disease in plants, (ii) to establish symbiotic relation with plants, and (iii) to support the catabolism of cellulose to produce energy” – what about saprotrophic fungi?
The sentence was modified to better highlight the function of cellulolytic enzymes in the recycling of dead plant material: "Across environments, beside supporting the catabolism of cellulose from live and dead plant material, cellulases produced by fungi are required to infest and cause disease in plants and to establish symbiotic relation with plants"
Line 67 – “However, these last few families are more frequently associated with other activities” – which activities?
Specific examples are being provided for each family (LL70-72).
Line 149 – “This suggests that cellulose deconstruction is a core function and that evolution towards increased cellulolytic potential is central to the evolution of fungi” – What about that parasitic organisms have decreasing genomes due to their increasing metabolic specialization? Could you refer your statements more to evolution data?
The paragraph (ll158-165) has been modified and references provided to better explained this concept.
Figures – Please correct axis names. I suppose that it should be for example “Number of genes coding for cellulases”. Please, also check across the manuscript (f. e. line 160).
The legend of the figure was modified for clarification.
Line 163 – “This suggests that some members of the Ascomycota and Basidiomycota phyla, with reduced number of predicted enzymes for cellulose deconstruction, are not actively degrading the plant cellulose.” – Are you sure? What about gene expression regulation? What about enzyme kinetics?
This is correct. The sentence was modified to highlight possible other mechanisms used to compensate for the reduced number of cellulolytic genes. The sentence now reads " This suggests that some members of the Ascomycota and Basidiomycota phyla, with reduced number of predicted enzymes for cellulose deconstruction, are not actively degrading the plant cellulose or use different mechanisms to produce cellulolytic enzymes (e.g., gene overexpression)."
Line 166 – “However, unlike for GHs, many genomes had only few genes for potential cellobiose dehydrogenases (i.e., AA3)” – or even one.
The sentence was reworded: However, unlike for GHs, many genomes had reduced number of genes for predicted cellobiose dehydrogenases.
Line 171 – “Having several -generally coregulated- genes for cellulases increases the overall cellulolytic activity.” - I suppose I know what do you mean, but I thing that these sentence needs rephrasing to clarify.
Additional references about cellulase regulation in fungi have been added.
Line 189 – Please, give full fungal names. Please, check across the manuscript.
The references for these genomes were added. However, I believe that providing the full name of all the sited genomes would be overwhelming and would make the reading difficult. Thus, as mentioned in a previous section (LL149-153) "All the abbreviation and enzyme name are consistent with the publicly accessible data. When a specific genome is mentioned, the corresponding reference is provided. However, readers are encouraged to check the MycoCosm portal to retrieve additional information and references."
Reviewer 3 Report
Berlemont presents an Entry entitled "The potential for cellulose deconstruction in fungal genomes". The author analyses fungal genomes currently available at MycoCosm database in terms of the presence of genes encoding cellulolytic enzymes. The entry gives a nice and comprehensive overview on potential enzymes and hightlights the importance of genome analysis.
The entry would benefit from some additional information on how data were extracted from the database.
Please revise the reference list according to the journals instruction (names of species in italics, check for capital letters)
Author Response
Berlemont presents an Entry entitled "The potential for cellulose deconstruction in fungal genomes". The author analyses fungal genomes currently available at MycoCosm database in terms of the presence of genes encoding cellulolytic enzymes. The entry gives a nice and comprehensive overview on potential enzymes and hightlights the importance of genome analysis.
The entry would benefit from some additional information on how data were extracted from the database.
The direct link to the data is in the text.
Please revise the reference list according to the journals instruction (names of species in italics, check for capital letters)
Thank you for pointing at this, the references were checked and the names of species are now in italic!
Reviewer 4 Report
- The work is well developed and extremely useful for those interested in the subject. I only recommend a couple of changes to clarify for readers who are new to bioinformatics and genomic databases.
Lines 32-35
In recent years, the high throughput DNA sequencing and ... enzymes and microbes [8–11].
I would add a few sentences explaining that the large number of sequenced fungal genomes (in whole or in part) is held in databases appropriate for such use, e.g., GenBank [xx] or Ensembl Genomes [Nucleic Acids Research, doi:10.1093/nar/ gkab1007] and a short introduction to MycoCosm
e.g.,
MycoCosm [xx], for example, is a genome portal developed by the US Department of Energy Joint Genome Institute to specifically support the integration and analysis of fungal genome sequences and other "omics" data.
line 76
Cellobiohydrolases (non-reducing end), aka, E.C.3.2.1.91, hydrolyze the glucosidic linkages in cellulose and cello-oligosaccharides from their non-reducing end and release cellobiose (G2). Known protein domains with Cellobiohydrolases activity are from the GH families 5, 6, and 9.
Cellobiohydrolases (non-reducing end)
Lines 134-136
Specifically, the Mycocosm project at the Joint Genome Institute (https://mycocosm.jgi.doe.gov/mycocosm/home)[11,27] aims to ...
This link is outdated. The JGI Genome Portal is now at https://genome.jgi.doe.gov/portal/.
Figure 1.
.the description does not fully correlate with the graphs presented.
Lines 105-109
Specifically, only ~60% of the characterized GH5 are described as cellulases. Similarly, ~75% of GH12, ~96% of GH9, and ~75% of the GH48 target the cellulose. The non-cellulolytic enzymes from these GH families include some GH5-mannases, some licheninases (GH5, GH9, GH12), some xylanases (GH5, GH8, GH12), and some chitinases (GH48), among others.
This observation. Does it come from the bibliography or from the analysis of the CAZy base?
Line 150
However, these trends were not supported across the genomes from the less abundant phylum Cryptomycota (n=2 genomes), Mucoromycota (n=24), Microsporidia (n=8) and 152 Zoopagomycota (n=7).
I would add that this observation needs to be confirmed when more species of these phyla are sequenced.
Author Response
The work is well developed and extremely useful for those interested in the subject. I only recommend a couple of changes to clarify for readers who are new to bioinformatics and genomic databases.
Lines 32-35 In recent years, the high throughput DNA sequencing and ... enzymes and microbes [8–11]. I would add a few sentences explaining that the large number of sequenced fungal genomes (in whole or in part) is held in databases appropriate for such use, e.g., GenBank [xx] or Ensembl Genomes [Nucleic Acids Research, doi:10.1093/nar/ gkab1007] and a short introduction to MycoCosm. e.g., MycoCosm [xx], for example, is a genome portal developed by the US Department of Energy Joint Genome Institute to specifically support the integration and analysis of fungal genome sequences and other "omics" data.
Modified as suggested.
line 76 - Cellobiohydrolases (non-reducing end), aka, E.C.3.2.1.91, hydrolyze the glucosidic linkages in cellulose and cello-oligosaccharides from their non-reducing end and release cellobiose (G2). Known protein domains with Cellobiohydrolases activity are from the GH families 5, 6, and 9. Cellobiohydrolases (non-reducing end)
I am unsure about this comment.
Lines 134-136 Specifically, the Mycocosm project at the Joint Genome Institute (https://mycocosm.jgi.doe.gov/mycocosm/home)[11,27] aims to ...
This link is outdated. The JGI Genome Portal is now at https://genome.jgi.doe.gov/portal/.
Thank you for pointing at this! The text has been modified as suggested and now correctly points to the active website for the MycoCosm portal
Figure 1..the description does not fully correlate with the graphs presented.
The legend was modified to better reflect the content of the figure.
Lines 105-109
Specifically, only ~60% of the characterized GH5 are described as cellulases. Similarly, ~75% of GH12, ~96% of GH9, and ~75% of the GH48 target the cellulose. The non-cellulolytic enzymes from these GH families include some GH5-mannases, some licheninases (GH5, GH9, GH12), some xylanases (GH5, GH8, GH12), and some chitinases (GH48), among others.
This observation. Does it come from the bibliography or from the analysis of the CAZy base?
It comes from a paper investigating the activity of enzymes reported on CAZy db. References are provided.
Line 150
However, these trends were not supported across the genomes from the less abundant phylum Cryptomycota (n=2 genomes), Mucoromycota (n=24), Microsporidia (n=8) and 152 Zoopagomycota (n=7).
I would add that this observation needs to be confirmed when more species of these phyla are sequenced.
Modified a suggested.
Round 2
Reviewer 2 Report
Accept
Reviewer 3 Report
Final proofreadings will deal with the spelling errors....
Reviewer 4 Report
The author has successfully responded to my corrections. I recommend that it be accepted in this form.